# CD3^+^ and CD8^+^ T-Cell-Based Immune Cell Score and PD-(L)1 Expression in Pulmonary Metastases of Microsatellite Stable Colorectal Cancer

**DOI:** 10.3390/cancers15010206

**Published:** 2022-12-29

**Authors:** Topias Karjula, Hanna Elomaa, Anne Niskakangas, Olli Mustonen, Iiris Puro, Teijo Kuopio, Maarit Ahtiainen, Jukka-Pekka Mecklin, Toni T. Seppälä, Erkki-Ville Wirta, Eero Sihvo, Juha P. Väyrynen, Fredrik Yannopoulos, Olli Helminen

**Affiliations:** 1Surgery Research Unit, Medical Research Center Oulu, Oulu University Hospital and University of Oulu, 90014 Oulu, Finland; 2Cancer and Translational Medicine Research Unit, Medical Research Center Oulu, Oulu University Hospital and University of Oulu, 90014 Oulu, Finland; 3Department of Biological and Environmental Science, University of Jyväskylä, 40014 Jyväskylä, Finland; 4Department of Education and Research, Central Finland Health Care District, 40620 Jyväskylä, Finland; 5Department of Pathology, Central Finland Health Care District, 40620 Jyväskylä, Finland; 6Faculty of Sport and Health Sciences, University of Jyväskylä, 40014 Jyväskylä, Finland; 7Faculty of Medicine and Health Technology, Tampere University and TAYS Cancer Center, Tampere University Hospital, 33520 Tampere, Finland; 8Department of Gastrointestinal Surgery, Helsinki University Central Hospital, University of Helsinki, 00290 Helsinki, Finland; 9Applied Tumour Genomics, Research Program Unit, University of Helsinki, 00290 Helsinki, Finland; 10Department of Gastroenterology and Alimentary Tract Surgery, Tampere University Hospital, 33520 Tampere, Finland; 11Central Hospital of Central Finland, 40014 Jyväskylä, Finland; 12Department of Cardiothoracic Surgery, Oulu University Hospital, 90014 Oulu, Finland

**Keywords:** colorectal carcinoma, pulmonary metastases, tumour-infiltrating lymphocytes, programmed death 1, programmed death ligand 1

## Abstract

**Simple Summary:**

The lung is the second most common site of metastases in colorectal cancer (CRC). The aim of our study was to evaluate the prognostic value of CD3^+^ and CD8^+^ T-cell density based immune cell score (ICS) and PD-1/PD-L1 expression in resected pulmonary metastases of microsatellite stable CRC. The T-cell infiltration was higher in the first pulmonary metastases compared to primary tumour. Pulmonary metastases with high ICS had improved survival compared to low ICS after adjusting for confounders. High tumour cell PD-L1 expression was associated with favourable prognosis. Our results might have clinical feasibility in planning future therapies.

**Abstract:**

The objective of this study was to evaluate the prognostic value of CD3^+^ and CD8^+^ based immune cell score (ICS), programmed death -1 (PD-1) and programmed death ligand -1 (PD-L1) in pulmonary metastases of proficient mismatch repair colorectal cancer (CRC) patients. A total of 101 pulmonary metastases and 62 primary CRC tumours were stained for CD3^+^, CD8^+^, PD-1 and PD-L1 expression. The prognostic value of ICS, PD-1/PD-L1 expression in 67 first pulmonary metastases and 61 primary CRC tumour was analysed. Comparative analysis was also performed between primary tumours and pulmonary metastases, as well as between T-cell densities and PD-1/PD-L1 expression. The 5-year overall survival rates of low, intermediate, and high ICS in pulmonary metastases were 10.0%, 25.5% and 47.0% (*p =* 0.046), respectively. Patients with high vs. low ICS in pulmonary metastases had a significantly better 5-year survival (adjusted HR 0.25, 95% CI 0.09–0.75, *p =* 0.013). High tumour cell PD-L1 expression in the pulmonary metastases was associated with improved survival (*p =* 0.024). Primary tumour CD8^+^ expression was significantly correlated with all T-cell densities in pulmonary metastases. Conclusion: The ICS evaluated from the resected pulmonary metastases of CRC showed significant prognostic value. High PD-L1 expression in pulmonary metastases is associated with favourable prognosis.

## 1. Introduction

Colorectal cancer (CRC) is one of the most common malignancies globally and the third leading cause of cancer death worldwide [1]. Approximately 10% of patients have synchronous pulmonary metastases and about 5% of patients have disease recurrence with pulmonary metastases at 5 years after treatment of primary CRC [2]. The 5-year overall survival of CRC in all stages is 63%, being only 14% in stage IV disease [3].

The immune system plays a pivotal role in cancer progression [4]. Immunoscore is a host immune response classification system based on CD3^+^ and CD8^+^ T-cell densities at the centre and the invasive margin of the tumour. Immunoscore has been shown to have independent prognostic value in CRC and is proposed to be included in CRC TNM-staging (TNMi) [5]. It is also thought to impact metastatic dissemination, as synchronous metastases in CRC are associated with lower Immunoscore values in the primary tumour [6]. The immune contexture has been studied in CRC liver metastases [7,8], and increasingly in pulmonary metastases [9], reporting prognostic value of tumour-infiltrating lymphocytes also in the metastases of CRC.

Programmed death 1-receptor (PD-1) and its ligand 1 (PD-L1) act as an inhibiting signalling pathway for immune response and has provided a major target for immune checkpoint inhibitor treatment in cancer [10]. It is proposed that the overexpression of PD-L1 by cancer cells causes the blockade of PD-1 positive T-cell effector functions and thus promotes cancer immune escape [11]. In CRC, there are discordant reports of the prognostic value of PD-1/PD-L1 expression in primary tumours [12,13,14,15] and in metastases [9,16,17]. A recent meta-analysis reported PD-L1 expression being a negative prognostic factor in primary CRC irrespective of DNA mismatch repair (MMR) status [18].

PD-1/PD-L1 expression is also a predictor for immune checkpoint inhibitor treatment response in several cancers [19]. In CRC, immune checkpoint inhibitor treatment has promising results in MMR deficient (dMMR) patients [20], while in MMR proficient (pMMR) CRC patients, the response rates are low [21,22]. Interestingly, in early-stage pMMR colon cancer patients, high CD8^+^PD-1^+^ T cell infiltration in preoperative biopsies was a predictive factor for neoadjuvant immune checkpoint inhibitor treatment response [23] suggesting that a possible sub-group for immune checkpoint inhibitor treatment in pMMR CRC patients might lie in those with high T-cell infiltration.

A better conceptualization of immune cell expression and PD-1/PD-L1 signalling in CRC progression is essential for further treatment strategy development. The primary aim of this study was to determine the prognostic value of CD3^+^ and CD8^+^ based immune cell score (ICS) and PD-1/PD-L1 expression in resected pulmonary metastases of pMMR CRC. The ICS and PD-l/PD-L1 expression patterns were also compared with those of primary tumours.

## 2. Material and Methods

### 2.1. Study Design

All patients with histologically confirmed pulmonary metastases from colorectal carcinoma operated in Oulu University Hospital and Central Finland Central Hospital during 2000–2020 were included. A total of 106 pulmonary metastasectomies were performed to 74 patients during the study period. Adequate samples and representative immunohistochemical staining for ICS evaluation where available from 101 pulmonary metastasectomies and 61 primary tumours. This was a population-based retrospective cohort study; the study hospitals were the only hospitals performing pulmonary metastasectomies in their geographical area.

### 2.2. Data Collection

Patients were identified using surgical registries and pathology reports. All relevant clinical data was retrospectively collected from electronic patient record systems or paper records (patients operated before 2007 in Central Finland Central Hospital) used in the study hospitals. Tumour classification was updated according to the American Joint Committee on Cancer (AJCC) 8th edition of tumour-node-metastasis (TNM) classification [24]. Survival data until 31.12.2021 was received from Statistics Finland. The follow-up data was 100% complete.

Prospectively collected diagnostic haematoxylin- and eosin-stained (HE) histological samples of the primary CRC tumours and the pulmonary metastases were collected from the pathology archives and viewed with a light microscope by a histopathologist. The most representative slide (with the deepest invasion depth for the primary tumours) was selected and digitalized using an Aperio digital scanner AT2 Console (Leica Biosystems Imaging Inc. Nussloch, Germany).

### 2.3. Tissue Microarrays

For MMR status and *BRAF* mutation status assessment tissue microarray (TMA) blocks were prepared from formalin-fixed paraffin-embedded tissue samples. Tissue cores with a diameter of 1 mm were punched out and were set into premade “recipient” paraffin blocks. The location of the punch was determined from the digitalized diagnostic HE-stained samples. A pathologist confirmed the location of the punches; the core sites were chosen to best represent overall tumour morphology while avoiding necrosis. We obtained 1–2 cores from the tumour centre and 1–2 cores from the invasive margin from both the metastases and the primary CRC tumour. The TMA was constructed using a TMA Master II tissue microarrayer (3DHistech Ltd., Budapest, Hungary). The TMA blocks were cut into 3.5µm-thick sections for further staining and analysis.

### 2.4. Immunohistochemistry

Additional 3.5µm-thick slides were cut from the original tumour blocks for immunohistochemical (IHC) staining for cytotoxic CD3^+^ and CD8^+^ cell density evaluation and PD-1/PD-L1 expression. Staining for CD3^+^ and CD8^+^ was conducted with anti-CD3^+^ (LN10, 1:200; Leica Biosystems, Newcastle, UK) and anti-CD8^+^ (SP16, 1:400; Thermo Scientific, Fremont, CA, USA) antibodies using a Lab Vision Autostainer 480 (ImmunoVision Technologies Inc., Brisbane, CA, USA). Signal visualization was performed using diaminobenzidine and sections were counterstained with haematoxylin. Slides were scanned and digitalized with a NanoZoomer-XR (Hamatsu Photonics, Hertfordshire, UK) at ×20 magnification.

IHC staining for PD-1 (The Human Genome Organization (HUGO) name PDCD1) and PD-L1 (HUGO name CD274) was performed with anti-PDCD1 (SP269, 1:50; Spring Bioscience, Pleasaton, CA, USA) and anti-CD274 (E1L3N, 1:100; Cell Signalling Technology, Danvers, MA, USA) antibodies using a BOND-III Stainer (Leica Biosystems, Buffalo Grove, IL, USA). Antigen retrieval was performed with Tris/EDTA (BOND ER solution 2, pH 9; Leica Biosystems). Samples were incubated in room temperature with diluted antibodies for 30 min in PD-L1 staining and 20 min in PD-1 staining procedures.

MMR status was evaluated from primary tumour samples only and it was conducted by IHC staining for expression of MLH1, MSH2, MSH6 and PMS2. Staining was applied on the TMA sections using BOND-III stainer and BOND Polymer Refine Detection (Leica Biosystems). Antigen retrieval was performed by incubation with Tris/EDTA buffer (BOND ER solution 2, Leica Biosystems; pH 9) at 100 °C for 30 min. The used antibody dilutions were 1:50 for MLH1 (Novocastra, Leica Biosystems, Nussloch, Germany; NCL-L-MLH-1), 1:50 for MSH2 (Calbiochem, San Diego, CA, USA; NA27), 1:150 for MSH-6 (Epitomics, Burlingame, CA, USA; AC-0047 EU) and 1:100 for PSM2 (BD Biosciences, Pharmingen, San Diego, CA, USA; 5564151). A 30 min incubation time was used for antibodies. Tissue samples that exhibited positive staining for all four markers were considered MMR proficient and samples that negative for at least one of the four markers were classified as MMR deficient. For *BRAF* V600E mutation status evaluation, IHC was performed on the TMA sections of both metastases and primary tumour samples with BenchMark XT immunostainer (Ventana Medical Systems, Tucson, AZ, USA) using *BRAF* V600E mutation-specific mouse monoclonal antibody (clone: VE1, Spring Bioscience, Pleasonton, CA, USA; dilution 1:400). An amplification was performed with OptiView Amplification Kit (Ventana). Positive staining indicated *BRAF* V600E mutation, while negative staining indicated wild-type *BRAF*.

### 2.5. Scoring

The CD3^+^ and CD8^+^ T-lymphocyte densities from the tumour centre and invasive margin of primary tumours and pulmonary metastases were determined form the digitalized whole section samples using QuPath [25]. The first resected pulmonary metastases were used for ICS scoring. In patients with multiple pulmonary metastases resected simultaneously, the metastasis for ICS scoring was picked randomly. Representative areas from the tumour centre and the invasive margin were selected, and the cell densities were enumerated. The width of the invasive margin was selected manually using an annotation brush of a width of 720 µm and was set 360 µm into the tumour tissue and 360 µm into the healthy tissue which is illustrated in Figure 1. The T-cell scoring was calculated using the consensus validation Immunoscore method previously suggested by Pagès et al. [5]. The lymphocyte densities in the invasive margin and tumour centre were converted into percentiles by comparing the density to the densities of all colon tumour or pulmonary metastasis samples, resulting in four percentile scores for each tumour (CD3^+^ and CD8^+^ densities in both invasive margin and tumour centre). The mean of the percentile scores was calculated. A three-tiered categorization was used as suggested before [5]: a mean percentile of ≤25% was scored as low, >25% and ≤70% was scored as intermediate, and >70% was scored as high.

The PD-L1 expression in tumour cells and in tumour infiltrating immune cells were evaluated separately throughout the CRC tumour and pulmonary metastases whole section IHC-stained slides. The number of positively stained tumour cells and/or tumour-infiltrating immune cells were calculated and proportions in relation to PD-L1 negative tumour cells and/or tumour-infiltrating immune cells evaluated as described by de Marchi et al. [26]. A cut-off value of ≥5% for high vs. low PD-L1 expression was used. PD-1 expression was evaluated separately from invasive margin and tumour centre from whole section slides by calculating PD1-positive immune cells per mm^2^ using QuPath. The cut-off value for PD-1 positivity was selected from the receiver operating characteristics (ROC) curve. The used cut-off values in the tumour centre and invasive margin were 49 cells/mm^2^ and 110 cells/mm^2^ in pulmonary metastases, and 7.5 cells/mm^2^ and 15 cells/mm^2^ in primary tumours, respectively.

### 2.6. Outcomes and Definitions

Charlson Comorbidity Index (CCI) was used for comorbidity classification [27]. The metastatic cancer under treatment was included as one comorbidity. Disease free interval (DFI) was defined as interval from surgery of primary CRC tumour to the date of first clinical or radiological relapse of disease. Pulmonary metastases which were detected under 6 months after primary cancer treatment were deemed as synchronous.

The primary outcome of the study was 5-year overall survival from the date of metastasectomy to death due to any cause before end of follow-up. Only 1 patient died of other cause than cancer, therefore cancer-specific survival was not analysed.

### 2.7. Statistical Analysis

A chi-square test was used for group comparison in categorical variables. One-way ANOVA and Kruskal–Wallis tests were used for continuous variable group comparison. A Mann–Whitney U -test was performed in continuous variable group comparison. Spearman correlation co-efficient was used to compare correlation between skewed continuous variables. A ROC curve analysis was used for additional cut-off determination. A Kaplan–Meier survival curve was constructed from first metastasectomy to death or end of follow-up to visualize survival up to 5 years after pulmonary metastasectomy. Log rank tests were used to compare statistical significance. The estimates for hazard ratios (HR) with 95% confidence intervals (CI) were calculated using Cox regression. For multivariate analysis of ICS in the pulmonary metastases, the Cox regression model was adjusted for sex (male or female), age (as continuous variable), CCI (1 or ≥2), neoadjuvant therapy (yes or no), number of pulmonary metastases at diagnosis (1, 2, or ≥3), synchronicity of pulmonary metastases (synchronous/metachronous) and former liver metastasectomy (yes or no). In the primary tumours, the multivariate analysis of ICS was adjusted for sex (male or female), age (as continuous variable), CCI (1 or ≥2), neoadjuvant therapy (yes or no), CRC stage (I–II, III, or IV), CRC location (rectum or colon) and CRC grade (1, 2 or 3). Stepwise methods were not used in the Cox multivariate analysis. Statistical analysis was performed using IBM SPSS Version 28 (IBM corp., Armonk, NY, USA).

### 2.8. Ethical Aspects

The Oulu University Hospital Ethics Committee approved the study (EETTMK 81/2008). Due to retrospective nature of this study the Finnish National Authority for Medicolegal Affairs (VALVIRA) waived the need for informed consent.

## 3. Results

### 3.1. Patient Characteristics

The whole cohort consisted of 106 pulmonary metastasectomies from 74 CRC patients. 32 pulmonary metastasectomies were re-metastasectomies and were performed to 21 patients. Adequate samples and representative immunostainings were available in 101 pulmonary metastases and 61 primary CRC tumours. The immune cell densities were scored from 67 primary pulmonary metastases samples and 61 primary CRC tumour samples.

In the final cohort of 67 patients with pulmonary metastases, the median age was 69 years (IQR 63–75) and 49.3% (33) of patients were male. At the time of primary cancer treatment, 5 patients (7.5%) had stage I CRC, 18 patients (26.9%) had a stage II CRC, 27 patients (38.8%) had a stage III CRC and 18 patients (26.9%) a stage IV CRC. The median DFI after primary CRC resection was 338 days. Of pulmonary metastasectomies, R0 resection was achieved in 94.0% (63) of cases, 4 cases were R1 resections. The median follow-up time was 25.9 months (IQR 18.8–47.5), ranging from 1 month to 209 months. The 5-year overall survival was 31.6%.

### 3.2. Immune Cell Score

In the pulmonary metastases, there were 10 patients (14.9%) with a low ICS, 40 patients (59.7%) with an intermediate ICS, and 17 patients (25.4%) with a high ICS. There was no statistically significant difference in clinical parameters between the ICS groups in pulmonary metastases (Table 1). Apart from tumour cell PD-L1 expression, ICS groups and PD-1/PD-L1 groups were significantly associated (Table 1). The median of tumour centre and invasive margin CD3^+^ and CD8^+^ densities had a strong positive correlation with immune cell PD-1 expression in the tumour centre (*r_s_* = 0.631; *p <* 0.001) and in the invasive margin (*r_s_* = 0.697; *p* < 0.001), and immune cell PD-L1 expression (*r_s_* = 0.405; *p* < 0.001). T-cell densities and tumour cell PD-L1 expression were not correlated (*r_s_* = 0.061; *p* = 0.627) (Appendix A).

In the primary CRC specimens, 13 patients (21.3%) had a low ICS, 35 patients (55.7%) had an intermediate ICS, and 14 patients (20.9%) a high ICS. Clinical parameters showed no difference between the primary ICS groups (Appendix A). The ICS groups between primary CRC and pulmonary metastases were suggestively associated (Table 1). In continuous variable comparison, the invasive margin and tumour centre CD3^+^ and CD8^+^ T-cell median densities were statistically significantly increased in the first pulmonary metastases compared to the primary tumour (Figure 2). In T-cell density correlation analysis, especially the CD8^+^ density in the invasive margin of the primary tumour had statistically significant moderate positive correlation with T-cell densities in the pulmonary metastases (*r_s_* = 0.354–0.406; *p =* 0.008–0.002; Appendix A). ICS groups and PD-1/PD-L1 groups were associated also in the primary tumours. The median of tumour centre and invasive margin CD3^+^ and CD8^+^ densities had strong positive correlation with immune cell PD-1 expression in tumour centre (*r_s_* = 0.449; *p* < 0.001) and invasive margin (*r_s_* = 0.595; *p* < 0.001), moderate positive correlation with immune cell PD-L1 expression (*r_s_* = 0.361; *p* = 0.005) but no significant correlation with tumour cell PD-L1 expression (*r_s_* = 0.221; *p* = 0.096) (Appendix A).

Of all patients, 2 patients had a mutated *BRAF* V600E in pulmonary metastases, which both had an intermediate ICS. MMR status was determined from all patients; all patients were pMMR.

### 3.3. Immune Cell Score and Survival

The 5-year survival of the first pulmonary metastasectomy stratified by low, intermediate, and high ICS of the pulmonary metastases are shown in Figure 3. The 5-year overall survival rates in ICS patient groups were 10.0%, 25.5% and 47.0% (*p =* 0.046) (Table 2). The overall 5-year survival was significantly better in the high and intermediate ICS groups compared to the low ICS group (high vs. low adjusted HR 0.25, 95% CI 0.09–0.75, *p =* 0.013; intermediate vs. low adjusted HR 0.31, 95% CI 0.13–0.74; *p =* 0.008; Table 3).

The ICS in primary CRC tumours showed no significant correlation in 5- or 10-year survival (Table 2 and Table 3; Appendix A).

### 3.4. PD-1/PD-L1 and Survival

The overall 5-year K-M survival curves of the PD-1 and PD-L1 levels in the pulmonary metastases and primary CRC tumours are shown in Figure 4 and Appendix A. PD-1 expression had suggestive prognostic value on 5-year survival in the invasive margin (*p =* 0.066) and in the tumour centre (*p =* 0.114) of the pulmonary metastases (Figure 4). High tumour cell PD-L1 expression in the pulmonary metastases was significantly associated with better survival (*p =* 0.024; Figure 4). Tumour cell PD-L1 expression was rare, observed in only 6.6% of metastases and 5.7% of primary tumours. Immune cell PD-L1 expression in pulmonary metastases or primary tumour had no significant survival effect. The PD-(L)1 status in the primary tumour samples had no statistically significant survival effect in our data (Appendix A).

In the sub-group analysis of ICS-high pulmonary metastases (*n* = 17), the high PD-l values were significantly associated with better survival in the invasive margin (*p* < 0.001) and suggestively in the tumour centre (*p* = 0.076; Appendix A). In lower ICS groups, PD-1 expression showed no prognostic value. High tumour cell PD-L1 expression had suggestive association with better survival in the intermediate (*p =* 0.154) and high ICS groups (*p =* 0.138; Appendix A), but not in the low ICS group.

### 3.5. Post Hoc Analysis of Immune Cell Score

Due to the small sample size of the study and the fact that the consensus validation Immunoscore is primarily constructed for non-metastatic CRC [5], an additional analysis was performed with a two-tier categorization with a cut-off value selected from the ROC curve of the previously mentioned mean percentile scores. The ROC-curve is illustrated in the Appendix A. In the pulmonary metastases, the mean percentile score value cut-off for high vs. low ICS was 65%. In this two-tier classification, the ICS of the pulmonary metastases had a significant prognostic effect on the 5-year overall survival of patients (low 18.2%, high 56.5%; *p =* 0.009; Appendix A).

## 4. Discussion

We analysed the prognostic effect of the ICS in resected CRC pulmonary metastases. The main finding of this study indicated that the ICS determined from the first resected CRC pulmonary metastasis was of significant prognostic value. High tumour cell PD-L1 expression and high PD-1-positive immune cell density in the pulmonary metastases were associated with favourable prognosis.

The immune cell contexture in metastatic CRC has been studied in a few studies [7,9,28], however, they primarily focus on liver metastases. These previously mentioned studies have demonstrated significant heterogeneity in the immune cell contexture in the CRC metastases, suggesting that the ICS in the least immune-infiltrated metastases has the best prognostic value. T-cell densities in randomly selected metastases and the mean T-cell density values of all the patients’ metastases were also used as basis of ICS calculation and showed a lesser prognostic value compared least immune-infiltrated metastases [7,9]. Additionally, the ICS construction in these afore mentioned articles is not performed according to the percentile-score based validation consensus Immunoscore method [5]. To the best of our knowledge, no analysis has been performed on the ICS in only the pulmonary metastasis of CRC. In our study, the patients with a high or intermediate ICS in the primary pulmonary metastases of CRC showed significantly better survival compared to patients with a low ICS in the three-tier classification. Using cut-offs 25% and 70%, the 5-year overall survival difference according to K-M log rank test was statistically significant. Nevertheless, it is possible that the cut-points determined for primary colorectal tumours are not optimal for metastatic CRC. In the post hoc survival analysis of ICS according to the cut-off selected by ROC-curves, the ICS high vs. low was associated with significant prognostic value. Taken together, these results suggest that the ICS constructed from the first pulmonary metastases of CRC has prognostic value, but the optimal cut-points need to be further validated. In the primary tumours, the evident selection bias probably explains the lack of prognostic value of ICS in our data. There was no reference data on the ICS evaluation of primary tumours in patients which have received pulmonary metastasectomy.

A few studies have demonstrated a significant increase in CD3^+^ and CD8^+^ cell densities in metastasis compared to the primary tumour [9,28]. We confirmed the result also in the first pulmonary metastasis, where CD3^+^ and CD8^+^ T cell densities were significantly higher compared to the primary tumour. There was also statistically significant correlation between T-cell densities in the primary tumour and first pulmonary metastases, especially between primary tumour CD8^+^ density in the invasive margin and T-cell densities of first pulmonary metastases. When comparing T-cell densities in the primary tumour to densities in all resected pulmonary metastases, the correlation was stronger. In categorical variable comparison, the association between primary tumour and first pulmonary metastases three-tier ICS groups was not statistically significant (*p =* 0.075). Considering this, and the earlier finding that stage IV primary CRC tumours with distant metastases had lower densities of T-cells in the primary tumours compared to early-stage CRC tumours [6], it might be that, after metastatic dissemination, the T-cell reaction subsides in the primary tumour while it accelerates in the metastases which might have future clinical implications.

PD-1 and PD-L1 are one of the most studied immune checkpoint molecules [18,19,29]. The prognostic value of PD-1/PD-L1 expression in CRC metastases has discordant results. Ahtiainen et al. reported immune cell PD-1 expression in liver and pulmonary metastases having no statistically significant survival effect on 10-year survival [9], however a protective trend of immune cell PD-1 expression can be seen in the K-M curves on 5 years of survival; immune cell PD-L1 expression similarly had no significant survival effect. Alterio et al. reported PD-1/PD-L1 expression in CRC liver metastases having no survival effect [16], however the PD-1 expression analysis was not performed on invasive margin and tumour centre separately. Takasu et al. reported immune cell PD-1 expression in CRC liver metastases having a positive effect on survival in a multivariate model, PD-L1 expression had a positive effect only in univariate analysis [17]. Additionally, Kim et al. performed a survival analysis on *BRAF*-mutated mCRC patients, where high PD-L1 expression and infiltration of CD8^+^ cells were associated with better prognosis [30]. According to our study on pMMR metastatic CRC patients, there is a significant favourable prognostic effect of tumour cell PD-L1 expression in the pulmonary metastases and a suggestive favourable prognostic effect of PD-1 expression in the invasive margin and tumour centre of pulmonary metastases; the immune cell PD-L1 values showed no survival effect. The association between PD-L1 expression and tumour infiltrating lymphocytes is demonstrated previously in primary CRC tumours [31] and recently in metastatic CRC [9]. Our study confirmed the correlation of the immune cell PD-L1 expression and also of PD-1 expression with mean T-cell densities and ICS in both primary tumours and pulmonary metastases. In the ICS subgroup analysis, patients with high ICS in the pulmonary metastases, PD-1 expression in the invasive margin was significantly associated with favourable prognosis, whereas tumour cell PD-L1 expression was suggestively associated with favourable prognosis. In lower ICS groups, PD-1/PD-L1 expression had worse prognostic value. Taken together, it appears that whereas high tumour cell PD-L1 expression in primary tumours predicts poor survival at least in early stage CRC, it has an opposite effect in CRC metastases, despite the observation that PD-L1 expression elevates in metastases in comparison to the primary tumour [32]. Additionally, the prognostic effect of also PD-1 expression in the CRC pulmonary metastases seems to be altered. These findings might be indicative for an altered role in PD-1/PD-L1 signalling in metastatic dissemination, which might also affect development of novel immunotherapy strategies and require further studies.

High PD-L1 expression is reported as a significant predictive biomarker for PD-1/PD-L1 blockade in several primary tumours [33]. In dMMR CRC, anti-PD-1/PD-L1 treatment is indicated in early to metastatic stages of the disease, as pointed out in the recent ESMO-guidelines [34,35]. In pMMR patients, immune checkpoint blockade response rates have remained low [21,22] and a possible subgroup benefitting from immunotherapy is yet to be proven. Immunotherapy on metachronous metastases is also an unknown research field. However, there are several recent case reports reporting immune checkpoint inhibitor treatment on high PD-L1 expressing pulmonary metastases having efficient treatment responses [36,37]. The significant prognostic effect of PD-L1 expression in pulmonary metastases in our study might have clinical implications in immune checkpoint blockade treatment. There might be implications also in other novel immunotherapeutic treatments such as cancer peptide-based vaccines [38]. Since monoclonal anti-PD-1/PD-L1 immunotherapy has confronted setbacks including limited response rates, toxicity complications and financial restrictions, other novel immunotherapeutic agents and combinatory therapeutic strategies have been developed [38,39]. Preclinical studies have demonstrated promising results in PD-1 and PD-L1 B-cell epitope vaccines, producing effective treatment responses superior to monoclonal anti-PD-1/PD-L1 immunotherapy in mice models [40,41]. Additionally, preclinical studies on non-human primates demonstrated PD-1 B-cell peptide cancer vaccination having a similar efficacy than monoclonal anti-PD-1/PD-L1 therapy however with lower rates of adverse side effects [42]. All in all, additional studies are needed to further the understanding of PD-1/PD-L1 signalling and immune host response in metastatic CRC to identify the possible patient groups that might benefit from immunotherapy.

There are several strengths in our study. To the best of our knowledge, this is the largest cohort of pulmonary metastases from CRC that had so far been evaluated for the ICS and PD-1/PD-L1 status. The dual-institutional basis of the study can be considered also as a strength. As a population-based study, the selection bias is minimal and restricted to surgical patient selection. Nevertheless, there might be some differences in the patient selection for pulmonary metastasectomy between the study hospitals, since in Oulu University Hospital district, the treatment and follow-up of primary CRC in under a third of patients has not occurred in our study hospital where the patient received pulmonary metastasectomy. The greatest limitation of the study is its small sample size, which naturally resulted into a relatively long study period and might produce confounders due to the improvement in diagnostics and treatment. Concerning analysis on PD-L1, the small number of patients with high PD-L1 expression (≥5%) is also a limitation in our study. Additional studies are required to validate the optimal cut-offs of PD-1/PD-L1 expression for possible clinical implications in immunotherapy. Additionally, some variation existed regarding neoadjuvant treatment, which can have significant effect on inflammatory response. This was, however, taken into account in the adjusted analysis.

## 5. Conclusions

This study concludes the ICS from the resected pulmonary metastases of CRC have prognostic value. There is significant correlation between the immune cell densities of the primary tumour and pulmonary metastases and the immune infiltration also significantly increases in the metastases compared to the primary tumour. High tumour cell PD-L1 expression in the pulmonary metastases had significant association with better 5-year overall survival. Additionally, PD-1 expression in the invasive margin of pulmonary metastases had suggestive prognostic value.

## Figures and Tables

**Figure 1 cancers-15-00206-f001:**
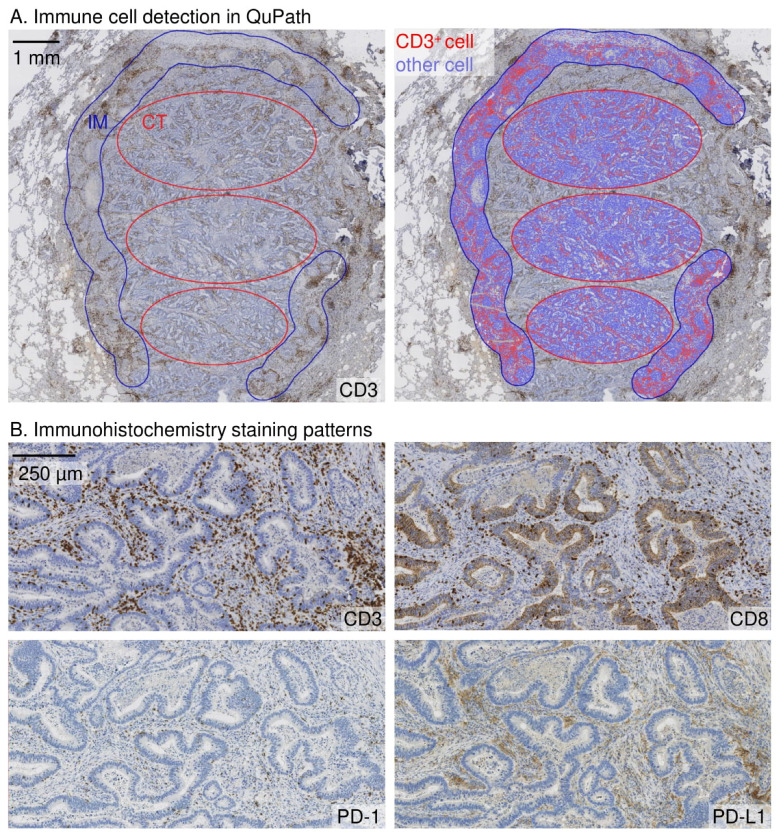
Immune cell density analysis and immunohistochemistry staining patterns in a pulmonary metastasis of colorectal cancer. (**A**). Analysis of immune cell density in the representative sites of the tumour centre (CT) and the invasive margin (IM) sites. The width of the invasive margin was 720 µm spanning 360 µm into the tumour and 360 μm into the healthy tissue. The immune cell density analyses for CD3, CD8 and PD-1 were done in QuPath bioimage software. PD-L1 expression was scored manually. (**B**). Examples of CD3, CD8, PD-1, and PD-L1 staining patterns are represented in the respective site of the tumour.

**Figure 2 cancers-15-00206-f002:**
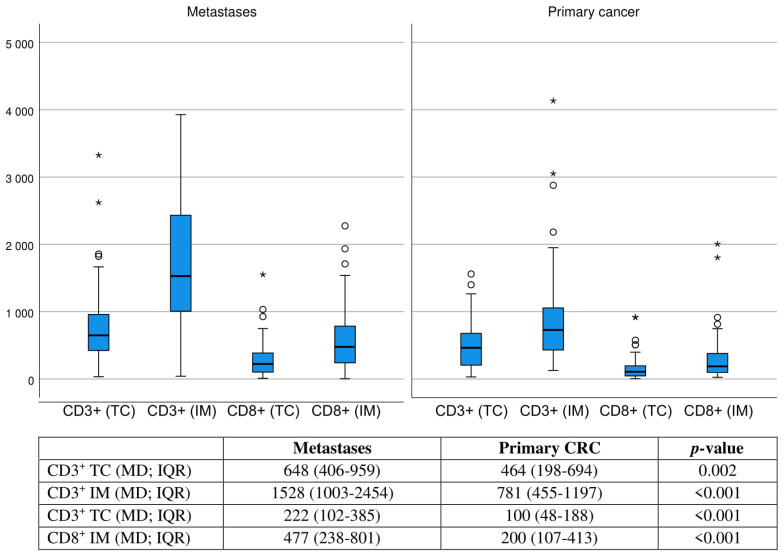
Boxplot of CD3^+^ and CD8^+^ densities (cells/1 mm^2^) at the tumour centre and invasive margin of first pulmonary metastases (*n* = 67) and primary tumour samples (*n* = 63). Mann–Whitney *U*-test was applied. Circles indicates outliers. Asterisks indicates extreme outliers.

**Figure 3 cancers-15-00206-f003:**
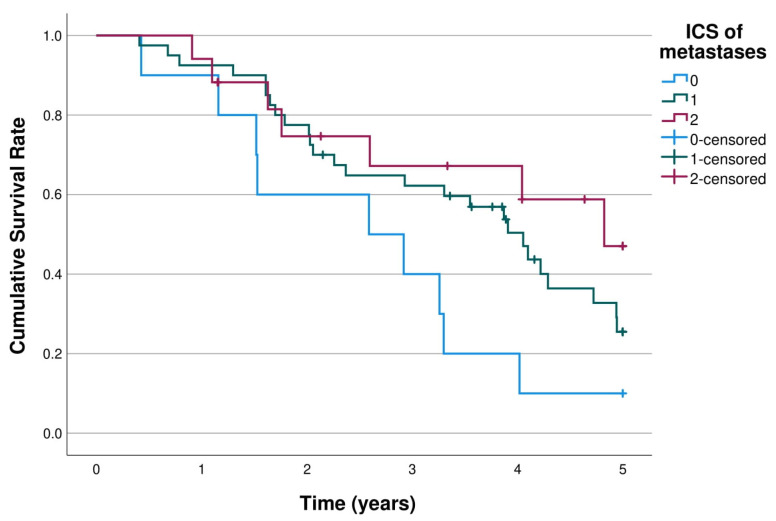
K m curves of 5-year survival of first pulmonary metastasectomy stratified by ICS of pulmonary metastases (*n* = 67). Log rank *p* = 0.046.

**Figure 4 cancers-15-00206-f004:**
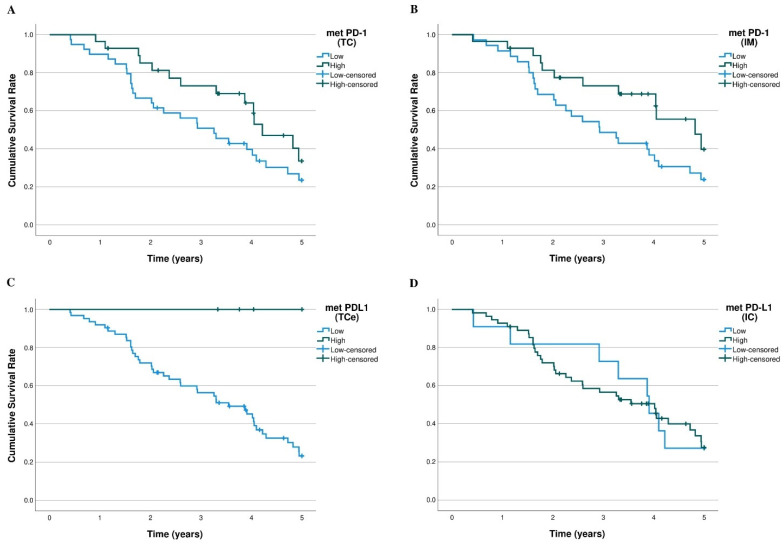
K-M 5-year overall survival curves of PD-1/PD-L1 expression in pulmonary metastases. (**A**) PD-1 expression (*n* = 67) in the tumour centre (*p =* 0.114). (**B**) PD-1 expression (*n* = 63) in the invasive margin (*p* = 0.066). (**C**) PD-L1 expression (*n* = 67) in the tumour cells (*p* = 0.024). (**D**) PD-L1 expression (*n* = 66) in the tumour infiltrating immune cells (*p =* 0.943).

**Table 1 cancers-15-00206-t001:** Baseline characteristics of pulmonary metastasectomy patients (*n* = 67) according to immune cell score from pulmonary metastases.

	ICS 1	ICS 1	ICS 2	*p*-Value
	*n* (%)	*n* (%)	*n* (%)	
	10	40	17	
Sex				0.615
Female	5 (50.0%)	22 (55.0%)	7 (41.2%)	
Male	5 (50.0%)	18 (45.0%)	10 (58.8%)	
Age (M, SD)	68.5 (10.5)	66.8 (11.3)	69.2 (9.6)	0.622
CCI				0.748
1	7 (70.0%)	25 (62.5%)	8 (47.1%)	
2	2 (20.0%)	8 (20.0%)	6 (35.3%)	
≥3	1 (10.0%)	7 (17.5%)	3 (17.6%)	
CRC stage				0.548
1–2	2 (20.0%)	17 (42.5%)	4 (23.5%)	
3	5 (50.0%)	13 (32.5%)	8 (47.1%)	
4	3 (30.0%)	10 (25.0%)	5 (29.4%)	
CRC location				0.255
Colon	6 (60.0%)	17 (42.5%)	11 (64.7%)	
Rectum	4 (40.0%)	23 (57.5%)	6 (35.3%)	
Former CRC liver metastasectomy				0.154
No	5 (50.0%)	25 (62.5%)	6 (35.3%)	
Yes	5 (50.0%)	15 (37.5%)	11 (64.7%)	
Neoadjuvant				0.883
No	5 (50.0%)	24 (60.0%)	10 (58.8%)	
Chemotherapy	5 (50.0%)	16 (40.0%)	7 (41.2%)	
DFI (d; MD; IQR)	645.5 (0–925)	363.5 (0–793)	309.0 (0–738)	0.922
Synchronicity				0.533
Synchronous	2 (20.0%)	10 (25.0%)	2 (11.8%)	
Metachronous	8 (80.0%)	30 (75.0%)	15 (88.2%)	
No of PM at diagnosis				0.369
1	5 (50.0%)	25 (62.5%)	13 (76.5%)	
≥1	5 (50.0%)	15 (37.5%)	4 (23.5%)	
Size of largest PM (MD; IQR)	2.8 (1.3–4.0)	2.2 (1.1–3.1)	2.0 (1.3–3.5)	0.691
Laterality of PM				0.392
Unilateral	8 (80.0%)	31 (77.5%)	16 (94.1%)	
Bilateral	2 (20.0%)	9 (22.5%)	1 (5.9%)	
*BRAF*				>0.999
Wild-type	10 (100.0%)	36 (94.7%)	17 (100.0%)	
Mutant	0 (0.0%)	2 (5.3%)	0 (0.0%)	
met PD-1 (TC)				<0.001 *
Low	10 (100.0%)	27 (67.5%)	2 (11.8%)	
High	0 (0.0%)	13 (32.5%)	15 (88.2%)	
met PD-1 (IM)				<0.001 *
Low	10 (100.0%)	23 (63.9%)	2 (11.8%)	
High	0 (0.0%)	13 (36.1%)	15 (88.2%)	
met PD-L1 (TCe)				0.156
Low	10 (100.0%)	38 (95.0%)	14 (82.4%)	
High	0 (0.0%)	2 (5.0%)	3 (17.6%)	
met PD-L1 (IC)				0.003 *
Low	5 (50.0%)	6 (15.4%)	0 (0.0%)	
High	5 (50.0%)	33 (84.6%)	17 (100.0%)	
prim PD-1 (TC)				0.008 *
Low	8 (88.9%)	19 (54.3%)	4 (25.0%)	
High	1 (11.1%)	17 (45.7%)	12 (75.0%)	
prim PD-1 (IM)				0.073
Low	7 (77.8%)	19 (54.3%)	5 (31.3%)	
High	2 (22.2%)	16 (45.7%)	11 (68.8%)	
prim PD-L1 (TCe)				0.074
Low	9 (100.0%)	34 (97.1%)	13 (81.3%)	
High	0 (0.0%)	1 (2.9%)	3 (18.7%)	
prim PD-L1 (IC)				>0.999
Low	7 (77.8%)	25 (71.4%)	11 (68.8%)	
High	2 (22.2%)	10 (28.6%)	5 (31.3%)	
ICS primary tumour				0.074
Low	4 (44.4%)	5 (13.9%)	4 (25.0%)	
Intermediate	5 (55.6%)	23 (63.9%)	6 (37.5%)	
High	0 (0.0%)	8 (22.2%)	6 (37.5%)	

CCI = Charlson comorbidity index; CRC = colorectal carcinoma; DFI = disease free interval; ICS = immune cell score; IC = tumour infiltrating immune cells; IM = invasive margin; IQR = interquartile range; met = metastases; PD-1 = programmed death 1; PD-L1 = programmed death ligand 1; prim = primary tumour; PM = pulmonary metastases; TC = tumour centre; TCe = tumour cells. Chi-square and Fisher’s exact tests were applied for categorical variables. Mann–Whitney U-test was applied for continuous variables. * Statistically significant at the 0.05 level (2-tailed).

**Table 2 cancers-15-00206-t002:** 5-year survival rates in pulmonary metastases and 10-year survival rates in primary colorectal tumours stratified by ICS and PD-1 and PD-L1.

	** *n* **	**ICS 0**	**ICS 1**	**ICS 2**	***p*-value**
Metastases	67	10.0%	25.5%	47.0%	0.046 *
Primary tumours	62	11.1%	41.9%	0.0%	0.152
	** *n* **	**PD-1 (TC) low**	**PD-1 (TC), high**		** *p* **
Metastases	67	23.5%	33.6%		0.114
Primary tumours	61	29.3%	27.2%		0.726
	** *n* **	**PD-1 (IM) low**	**PD-1 (IM), high**		** *p* **
Metastases	63	23.8%	39.7%		0.066
Primary tumours	60	20.7%	42.5%		0.328
	** *n* **	**PD-L1 (TCe) low**	**PD-L1 (TCe), high**		** *p* **
Metastases	67	23.3%	100.0%		0.024
Primary tumours	60	28.2%	0.0%		0.328
	** *n* **	**PD-L1 (IC) low**	**PD-L1 (IC), high**		** *p* **
Metastases	66	27.3%	27.7%		0.943
Primary tumours	60	32.4%	13.9%		0.086

* Statistically significant at the 0.05 level.

**Table 3 cancers-15-00206-t003:** Hazard ratios (HR) for 5-year all-cause mortality with 95% confidence intervals in pulmonary metastases and primary colorectal tumours stratified by ICS (0/1/2).

	*n*	ICS 0, HR (95%CI)	ICS 1, HR (95%CI)	ICS 2, HR (95%CI)
**Metastases**				
Crude	67	1.00 (reference)	0.48 (0.22–1.03, *p* = 0.058)	0.31 (0.12–0.84, *p* = 0.022)
Adjusted *	67	1.00 (reference)	0.31 (0.13–0.74, *p* = 0.008)	0.25 (0.09–0.75, *p* = 0.013)
**Primary Tumours**				
Crude	62	1.00 (reference)	0.63 (0.29–1.35, *p* = 0.233)	1.31 (0.53–3.24, *p* = 0.558)
Adjusted **	62	1.00 (reference)	0.69 (0.28–1.67, *p* = 0.406)	1.35 (0.41–4.42, *p* = 0.625)

* Adjusted for sex (male/female), age (continuous), CCI (1/≥2), Neoadjuvant chemotherapy (no/yes), number of pulmonary metastases at diagnosis (1/2/≥3) and synchronicity of first resected pulmonary metastases (synchronous/metachronous), former liver metastasectomy (yes/no). ** Adjusted for sex(male/female), age (continuous), CCI (1/2/≥3), Neoadjuvant therapy (no/yes), CRC stage (I-II/III/IV), CRC location (colon/rectum) and CRC grade (1/2/3).

## Data Availability

Data are available from the corresponding author upon reasonable request. Data sharing will require additional ethical board statement.

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
