# Peer review of "CD3+ and CD8+ T-Cell-Based Immune Cell Score and PD-(L)1 Expression in Pulmonary Metastases of Microsatellite Stable Colorectal Cancer"

_cancers, 2022, doi:10.3390/cancers15010206_

Round 1

Reviewer 1 Report

The manuscript titled “CD3+ and CD8+ T cell based immune cell score and PD-(L)1 expression in pulmonary metastases of microsatellite stable colorectal cancer” describes prognostic benefit from ICS and PD-1 level is associated with better overall outcomes of patients which might suggest novel strategies of PD-1/PD-L1 based immunotherapy. Overall, the manuscript is well-written and prepared. The followings are some concerns and comments have been pointed out that the authors may want to consider.

1) Page 2, paragraph 1 line 4: Please cite reference within the sentence. Check throughout the manuscript.

2) Page 3, 2.2 section paragraph 1 the last second line: It should be “30-day”.

3) Page 9, table 1 note: Please include statistical method.

4) Page 9, Figure 2 legend: Please include the meaning of “star”.

5) Page 10, Figure 3: Please include sample size.

6) Page 10, Table 2: Please use italic p and be consistent throughout the manuscript.

7) Page 10, Table 3: Please be consistent throughout the manuscript with “0.01” or “.01”.

8) Page 12, Figure 4: Please include sample size.

9) The authors suggested that PD-1 expression indicates better OS. I’d suggest the authors discuss PD-1/PD-L1 based immunotherapy with related vaccines and/or other small molecules which might lead to novel anti-cancer strategies. For example, PMID: 35646678; PMID: 36211807; PMID: 31177328.

Author Response

Thank you for the revision. All points are considered and applied in the new version of the manuscript. 

Additionally to these comments, we updated the survival data on our cohort, which shifted the results in this study. Changes on survival data were made accordingly on the survival effect on ICS and PD(L)1. Also further discussion on PD(L)1 based immunotherapy was added. 

Reviewer 2 Report

The study by Helminen et al, titled "CD3+ and CD8+ T cell based immune cell score and PD-(L)1 expression in pulmonary metastases of microsatellite stable colorectal cancer," examined the potential predictive value of T cells and immune checkpoint expression in patients with colorectal cancer that had spread to the lungs. The researchers found that the infiltration of T cells was higher in the first pulmonary metastases compared to the primary tumor and that those with high immune cell scores had improved survival compared to those with low scores after adjusting for other factors. High expression of PD-1 was also associated with a favorable prognosis. The observation is very interesting with potential clinical feasibility.

However, the study lacks mechanistic data or discussion and does not provide a rationale for why CD3+CD8+ T cells were chosen over other types of T cells, such as CD4+ T helper cells or Foxp3+ Treg cells. Additionally, the study only discusses PD-1 and PD-L1, but not other immune checkpoints.

The conclusions drawn from this study need to be tempered. For example, PD-L1 is expressed in both immune cells and tumor cells, the authors did not demonstrate in which cell type the PD-L1 is upregulated by cell type markers. When the author call “tumor cell PD-L1 expression” and “immune cell PD-L1 expression”, does it mean cell type or CT/IM location? as we know immune cells usually infiltrates TME and I can hardly tell the origin of PD-L1 from IHC in Fig.1.

Some minor concerns with the manuscript include formatting issues in the citations, please double-check. For example [16]. and the need for more background/mechanistic information and relevant references.

Author Response

Thank you for the revision. The CD3+ and CD8+ cells were picked for analysis in our study as they are the basis of Immunoscore classification. As reviewed in our article, the prognostic value on Immunoscore in CRC has been proven in large validation studies, and our study was planned as an extension to earlier studies on CD3+ and CD8+ based Immunoscore.

Also PD-(L)1 was picked for analysis for this signalling pathway is the most common immune checkpoint molecules and the only pathway on with immune checkpoint inhibitors are approved in guidelines. This rationale is added on the manuscript. 

Concerning PD-L1 expression, the analysis was performed on immune cells and tumour cells separately as described in the methods section. TC/IM refers to tumour centre and invasive margin, and this division concerns only PD-1 expression. 

The citations were corrected according to the author instructions. 

In addition to these comments, we updated the survival data on our cohort, which shifted the results in our article. Some minor changes have been made along the article concerning the survival of ICS and PD(L)1 expression. Also, further discussion was added on PD-(L)1 results as requested by reviewer 1.